# Burden of tuberculosis in underserved populations in South Africa: A systematic review and meta-analysis

**Lydia M. L. Holtgrewe** [1]*, **Ann Johnson**[1,2], **Kate Nyhan**[3,4], **Jody Boffa**[5], **Sheela V. Shenoi** [1,2], **Aaron S. Karat** [6], **J. Lucian Davis** [1,2☉], **Salome Charalambous**[1,5☉]

**1** Department of Epidemiology of Microbial Diseases, Yale School of Public Health, New Haven, Connecticut, United States of America, **2** Yale School of Medicine, New Haven, Connecticut, United States of America, **3** Harvey Cushing / John Hay Whitney Medical Library, Yale University, New Haven, Connecticut, United States of America, **4** Department of Environmental Health Sciences, Yale School of Public Health, New Haven, Connecticut, United States of America, **5** The Aurum Institute, Johannesburg, Gauteng Province, South Africa, **6** Tuberculosis Centre, London School of Hygiene & Tropical Medicine, City of London, United Kingdom

☉ These authors contributed equally to this work.

* lmlh1g18@soton.ac.uk

**Data Availability Statement:** All relevant data has been included in the article, appendix, or the supplementary materials. Additional datasets and

## Author summary

Identifying case-finding strategies to reduce tuberculosis (TB) incidence in high-burden countries requires better knowledge of the disease burden in key contributing populations and settings. To inform South Africa's National Tuberculosis Strategic Plan 2023–2028, we conducted a systematic review of active TB disease and latent TB infection (LTBI) prevalence and incidence in underserved populations, defined as those living in informal settlements, townships, or impoverished communities. We identified articles published from January 2010 to December 2023, assessed study quality, and conducted a meta-analysis to estimate pooled TB and LTBI prevalence stratified by HIV status. We calculated prevalence ratios for underserved populations compared to the overall South African population. The search yielded 726 unique citations. We identified 22 studies reporting TB prevalence (n = 12), TB incidence (n = 5), LTBI prevalence (n = 5), and/or LTBI incidence (n = 2) eligible for the review, including six high-quality studies. Meta-analysis demonstrated a high prevalence of TB disease among persons living without HIV (2.7% 95% CI 0.1 to 8.5%) and persons living with HIV (PLWH) (22.7%, 95% CI 15.8 to 30.4%), but heterogeneity was high ($I^2$ = 95.5% and 92.3%, p-value<0.00). LTBI prevalence was high among persons living without HIV (44.8%, 95% 42.5 to 47%) with moderate heterogeneity ($I^2$ = 14.6%, p-value = 0.31), and lower among PLWH (33%, 95% CI 22.6 to 44.4%) based on one study. Compared to the national average, underserved populations of persons living without HIV had a 4-fold higher TB prevalence and a 3.3-fold higher LTBI prevalence. Underserved PLWH had a 31-fold higher TB prevalence than the national average, but similar LTBI prevalence as measured in one study. Our findings illustrate that underserved populations in South Africa have a substantially higher TB and LTBI prevalence than the general population, making targeted TB interventions potentially beneficial. More research is needed to explore the heterogeneous TB epidemiology in South Africa.

analytical code can be accessed under https://osf.io/uzj65/?view_only=e562685a34804ebb8a8f0286804ae4ce.

**Funding:** The authors received no specific funding for this work.

**Competing interests:** The authors have declared that no competing interests exist.

## Introduction

The 2023 WHO Global Tuberculosis (TB) Report found a significant global recovery in the number of people diagnosed with and treated for TB, following two years of disruptions to TB services due to the COVID-19 pandemic [1]. In spite of this, TB remains the world's second leading infectious disease killer after COVID-19 [1]. The 2015 End TB Strategy proposed ambitious targets to reduce TB incidence by 80% and TB deaths by 90% by 2030 [2]. While cumulative global reductions in these indicators fell short of the 2020 interim milestones, South Africa reported a 53% reduction in incidence between 2015 and 2022, from 988 to 468 persons with TB per 100,000 population, although the TB mortality has not fallen as rapidly [1–4]. Novel strategies are needed to reach the WHO targets.

Addressing the diagnostic gap in South Africa and other high-burden countries demands current and reliable TB prevalence and incidence estimates, especially among key populations at highest risk of TB [5]. Such estimates can be obtained using mathematical modelling or population-based studies. The former use a variety of readily available data inputs, including annual notifications of people with TB and previous surveys of TB prevalence and TB risk factors to estimate TB incidence [6]. Although modelling studies are convenient and less costly than population-based surveys, it can be challenging to obtain reliable sub-national estimates of TB incidence and prevalence given local heterogeneity in notifications of people with TB and in prevalence of HIV and other TB risk factors [7–10]. Therefore, the WHO recommends using population-based studies to obtain reliable estimates that can be used by policymakers to prioritise TB service delivery for those at highest risk and project the added impact of targeted interventions [11].

One key population of particular interest is people who are 'underserved', a group who faces structural barriers to accessing TB services because of disadvantaged or marginalised socioeconomic positions that require them to live in areas with fewer nearby clinics [12]. In the context of South Africa's social system, we defined underserved populations as those living in informal settlements, townships, or other impoverished communities, in line with the United Nations' third Sustainable Development Goal [13, 14]. Underserved populations share individual and environmental risk factors for acquiring TB infection and/or progressing to active TB disease, such as HIV, malnutrition, diabetes, overcrowding, poor ventilation, urban residence, and poor access to health services [15–17]. In a previous scoping review, we found populations living in informal settlements to be the largest contributor to the absolute number of people with TB in South Africa followed by people living with HIV (PLWH) [14]. To inform South Africa's revised National TB Strategic Plan 2023–2028 [5], we undertook an updated systematic review to determine the prevalence and incidence of active and latent TB disease among underserved populations in South Africa between 2010 and 2023.

## Methods

### Study protocol and search strategy

We developed a systematic review protocol following the PRISMA-P reporting guidelines [18] (see *S1 Checklist*) for systematic reviews and prospectively registered it with PROSPERO [19]. Our search strategy identified published literature and pre-prints in the databases *Lens.org*, *EMBASE*, *Africa Index Medicus* and the Clarivate *Incidence & Prevalence Database* on June 26, 2023. *Lens.org* aggregates content from multiple sources including *PubMed*, *Microsoft Academic*, and *Crossref*, and helps identify free full text versions of papers. Our search strategy consisted of key words, database-specific subject headings, and title/abstract search terms (see *S1 Table*). We combined terms related to active TB disease, latent TB infection (LTBI),

epidemiologic measures of disease burden, South Africa, and underserved populations. We also searched journal names and full text articles for terms related to South Africa to capture articles whose titles and abstracts may have omitted this information. Finally, we conducted a backward search of the reference list for our prior systematic review on TB incidence and prevalence in informal settlements in South Africa [15].

## Study selection

We imported search results into Covidence systematic review software (Melbourne, Australia) and removed all duplicates. Two reviewers (L.H., A.J.) independently conducted title and abstract screening, followed by full-text screening. We included 1) peer-reviewed articles and pre-print manuscripts of prospective and retrospective cohort studies, cross-sectional studies, non-randomized studies and randomized controlled trials that were 2) published or made available between 1 January 2010 and 26 July 2023, regardless of language. We required studies to 3) report prevalence, incidence, or notification data on active TB disease or LTBI, and to include 4) underserved populations in South Africa, including townships, informal settlements, and impoverished populations. To guide screening of articles, we defined townships as tightly regulated, racially segregated residential areas built outside cities during the Apartheid Era. We defined informal settlements as residential areas constructed on land that occupants have no legal claim to occupy, often in the context of rapid population growth and an inadequate housing supply. Both are characterized by substandard living conditions [20, 21]. We defined impoverished communities as those consisting of individuals of low socio-economic status, characterised by low household income [22]. Because studies reported wealth and income in different ways, we classified a study population as impoverished if the study referenced any measure or proxy indicative of low household wealth or income. We resolved discrepancies between reviewers on study inclusion or exclusion by consensus.

## Data extraction

One reviewer (L.H.) collected relevant study characteristics using standardised data extraction forms, including details on study design, TB diagnostic tools, and participant demographic and clinical characteristics (see *S2 Table*). We recorded prevalence and cumulative incidence as the number of people with TB per 100,000 and incidence rates as the number of people with TB per 100,000 person-years, including 95% confidence intervals (CI) if available. If the estimates were not directly reported, we calculated them by dividing TB notifications by the population size (prevalence) or total population at risk (cumulative incidence). We stratified all outcome estimates by HIV status (PLWH and people living without HIV). When study outcomes were not broken down by HIV status, we reported the overall study cohort's outcomes (people living with and without HIV). A second reviewer (A.J.) checked the extracted data for accuracy.

## Risk-of-bias assessment

Two reviewers (L.H., A.J.) independently assessed for risk-of-bias among included studies using all nine items in *JBI's Prevalence Critical Appraisal Tool* using standardised forms (see *S3 Table*). Items 1–5 evaluate for selection bias and generalisability by assessing the sample frame; sampling approach; sample size; participant characteristics and study setting; and sample coverage to determine if the study population is representative of the target population and sufficiently large. Items 6–7 address measurement error, including whether valid methods were used to identify the condition and applied consistently to all participants. Last, items 8–9 examine the statistical methods used, including the appropriateness of the statistical analysis

plan and the adequacy of the response rate [23]. We scored each item as 'Yes', 'No', 'Unclear' or 'Not applicable', resulting in an overall decision to either label studies as 'low risk-of-bias' or 'high risk-of-bias'. A study's risk of bias refers to the potential influence of its methods on the observed outcomes. Studies classified as having a high risk-of-bias are more likely to deviate in their estimates of the true effect than those classified as having a low risk-of-bias. Because of the importance of sample frame (item 1), sampling approach (item 2), and diagnostic methods (item 6) for determining prevalence and incidence in a target population, studies receiving a 'No" response to any of these items were labelled as 'high risk-of-bias'. Reviewers resolved all discrepancies through discussion.

## Data synthesis

We summarised the characteristics and outcomes of all individual studies descriptively, then undertook meta-analyses of the pooled prevalence of active TB disease, prevalence of LTBI, incidence of active TB disease, and incidence of LTBI, reported as standardised proportions, and stratified by HIV status. For the summary estimates of the burden of active TB disease, we only included studies using WHO-endorsed diagnostic tools [19, 24–26], including mycobacterial culture, smear microscopy, loop-mediated isothermal amplification (LAMP)-based assays, automated nucleic acid amplification tests (NAAT) and lateral flow lipoarabinomannan assays (LF-LAM) for TB diagnosis in the entire study cohort [24]. For summary estimates of the burden of LTBI, we included studies using tuberculin skin tests (TST) or interferon-γ-release assays (IGRA) for LTBI diagnosis in the entire study cohort. The criteria for conducting meta-analysis required the presence of two or more studies within each meta-analysis group, provided that those studies reported sufficiently similar summary estimates using the same prevalence or incidence units to ensure comparability.

## Primary analyses

We conducted the meta-analysis according to the recommendations of the *Cochrane Handbook for Systematic Reviews* [26]. Specifically, we conducted a proportion meta-analysis after recasting outcome values using the double arcsine transformation to stabilise variances and ensure interpretable confidence intervals [26, 27]. We fitted DerSimonian and Laird inverse-variance random-effects models given the statistical heterogeneity among studies and their clinical and methodological variability [28]. We explored statistical, methodological, and clinical heterogeneity through visual inspection of forest plots and the $I^2$ statistic [29]. We assessed for publication bias using funnel plots and the Egger test at a significance level of $\alpha = 0.05$.

## Sensitivity and subgroup analyses

We repeated the meta-analysis after excluding studies that were statistical or clinical outliers (e.g., different population characteristics or study settings) within each HIV status subgroup (*see S1 Data*) [26]. We also examined the influence of diagnostic tools, age, sex, ethnicity, study setting, and studies' risk of bias on outcome estimates.

## Other effect estimates

We calculated the prevalence ratio (PR) for active TB disease among underserved populations by dividing pooled summary estimates, obtained through sensitivity analysis, by the most recent national TB prevalence figures from South Africa's 2018 national TB prevalence survey (see *S2 Data*) [30]. We calculated the prevalence ratio for LTBI in underserved populations by using modelled LTBI prevalence estimates from a 2014 study of the global burden of LTBI as

the denominator (see *S2 Data*) [31]. We conducted all analyses in *R Core Team (2023)* using '*meta*' [32], '*metafor*'[33] and *Wang*'s R code [34]. The analytic code used to generate meta-analyses can be accessed under https://osf.io/uzj65/?view_only= e562685a34804ebb8a8f0286804ae4ce.

## Results

### Search process and study selection

The search returned 1426 search results (see *Fig 1*). After excluding 700 duplicates, we screened 726 titles and abstracts. All screened articles can be found in *S3 Data*. We selected 74 reports for full-text review, of which 21 studies met our inclusion criteria. One study included in the previous version of the review [14] met out inclusion criteria and was added, resulting in a total of 22 studies in this review [35–56]. The most common reasons for exclusion were ineligible study design and setting, not including underserved populations, and/or ineligible study outcomes like TB morbidity, TB mortality, and TB-related treatment costs.

### Study characteristics

Out of the 22 studies included in this study, 17 studies provided prevalence estimates [35–51], while six studies reported incidence figures [49, 52–56] (Table 1). Notably, only three [38, 39, 52] of these 22 studies were also captured in the previously conducted scoping review on TB prevalence and incidence in informal settlements in South Africa [14]. Study participants were recruited from outpatient facilities (n = 13) [35, 37–40, 48, 49, 51–55], homes (n = 9) [41–47, 50, 56] and a hospital (n = 1) [36]. Most studies (n = 16) [35–41, 43, 44, 48–54] were conducted in townships in the Western Cape Province. Other study locations included Gauteng [42, 46, 47, 56], Eastern Cape [45] and KwaZulu-Natal [55] provinces. Bacterial culture was the most frequently used active TB diagnostic tool of active TB disease (n = 6) [35, 37–40, 43], followed by self-report (n = 4) [44–46, 56]. Other studies used multiple methods (n = 7) [36, 41, 42, 52–55], including combinations of bacterial culture, NAAT, sputum smear microscopy, and clinical scoring systems. TST was the most commonly used diagnostic tool for LTBI (n = 6) [47–50, 53], with a positivity threshold of ≥10 mm among people living without HIV and ≥5 mm among PLWH. One study used IGRA [51]. Several studies recruited targeted rather than population-based samples, including eight studies in which the majority of participants were women [35–39, 47, 52, 55] and six studies focused on PLWH only [35–38, 52, 55].

### Risk-of-bias assessment

As shown in *Table 2*, six studies were labelled as low risk-of-bias, including one study reporting active TB disease prevalence [41], three reporting LTBI prevalence [47–49], one reporting active TB disease incidence [52], and one reporting LTBI incidence [49]. The most common reasons for high risk-of-bias scores were methodological weaknesses in sampling, outcome ascertainment, and statistical analysis. Sampling weaknesses included recruitment from non-representative populations, including six studies enrolling only PLWH and one enrolling only hospitalised individuals; low response rates from men in eight studies, resulting in female-predominant samples; and non-random sampling, such as convenience sampling in eleven studies. In five studies, outcome ascertainment was limited by use of non-standardised or subjective diagnostic tools, such as self-report or clinical scoring systems. Last, statistical analyses lacked sample size calculations in 22 studies, while nine studies presented incidence or prevalence estimates without confidence intervals. Please refer to *Tables A and B in S4 Table* for a more detailed justification of the overall appraisal of individual studies.

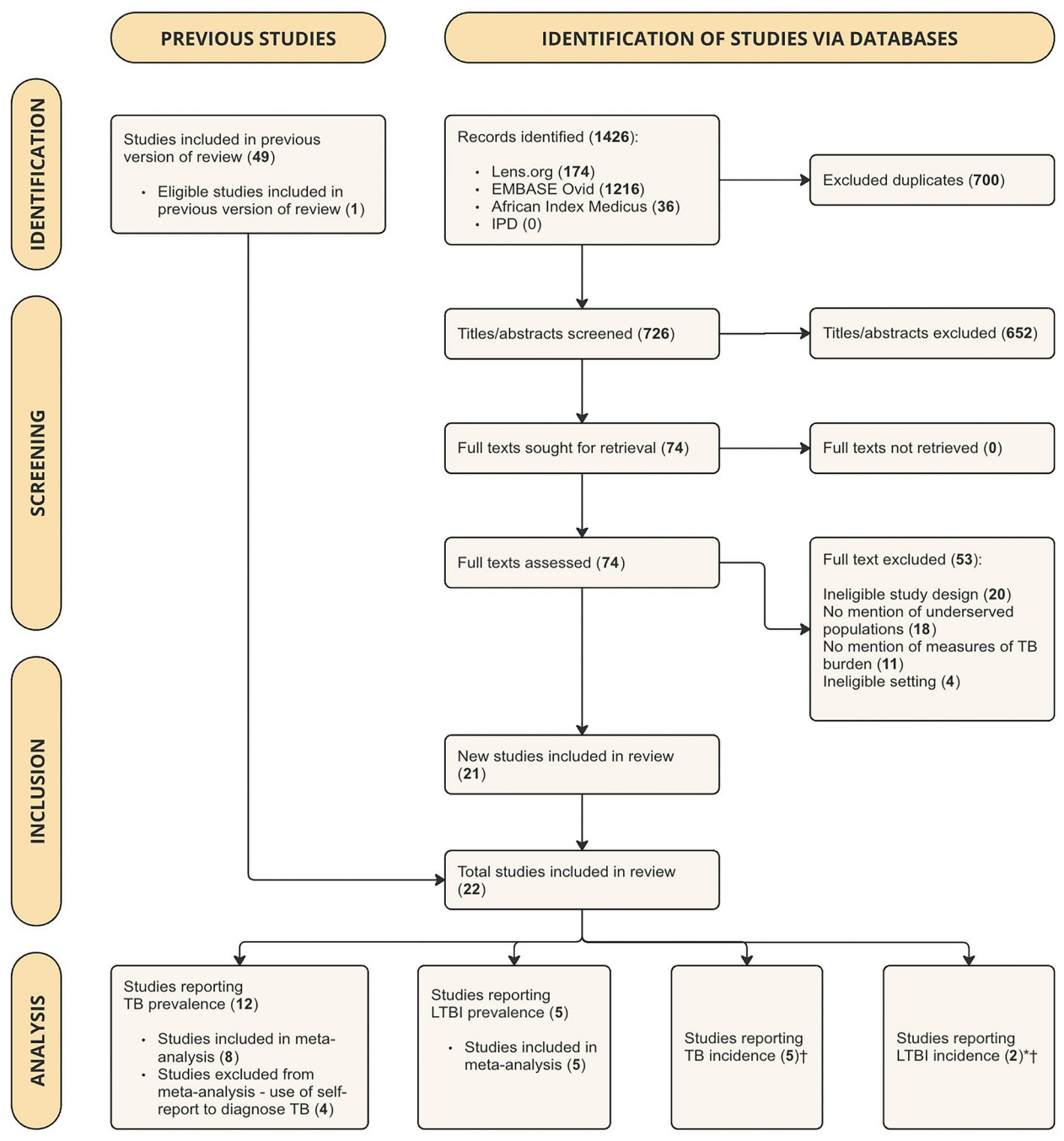

**Fig 1. PRISMA flow diagram, showing the results of study search and screening procedures. Abbreviations**: IPD = Incidence and Prevalence Database (Clarivate); TB = Tuberculosis; LTBI = Latent Tuberculosis. *Because Martinez et al. (2017) reports both TB incidence and LTBI incidence data, and Middelkoop et al. (2014) reports both LTBI prevalence and LTBI incidence data, these studies were counted twice in the analysis. †Because of the insufficient number of eligible studies (<2 in each meta-analysis) that reported summary estimates using consistent incidence units, we did not conduct meta-analyses for TB and LTBI incidence.

## Results of individual studies

**Prevalence of active TB disease and LTBI.**    Twelve studies reported an active TB disease prevalence estimate in adults, ranging from 0.4% [42] to 34% [40] (Table 3). Overall, studies

**Table 1. Characteristics of included studies.**

| Data extractors (Date) | Author (Year) | Study design (Study setting) | Sampling strategy | Diagnostic tool | Exposure definition | Geographic location | Sample size | Age group | Female (%) | PWH (%) |
|---|---|---|---|---|---|---|---|---|---|---|
| **a. TB PREVALENCE** | | | | | | | | | | |
| L.H.; A.J. (01.03.2023) | Lawn (2011) [35] | Cross-sectional (Outpatient) | Convenience sampling | Bacterial culture | Township population | Gugulethu Township, Cape Town | 468 | Adults | 66 | 100 |
| L.H.; A.J. (04.03.2023) | Lawn (2017) [36] | Cross-sectional (Hospital) | Convenience sampling | Automated NAAT; Bacterial culture | Township population | G.F. Jooste Hospital, Cape Town | 427 | Adults | 61 | 100 |
| L.H.; A.J. (07.03.2023) | Lawn (2011) [37] | Cross-sectional (Outpatient) | Convenience sampling | Bacterial culture | Township population | Gugulethu Township, Cape Town | 542 | Adults | 64 | 100 |
| L.H.; A.J. (21.03.2023) | Dawson (2010) [38] | Cross-sectional (Outpatient) | Convenience sampling | Bacterial culture | Township population | Gugulethu Township, Cape Town | 235 | Adults | 73 | 100 |
| L.H.; A.J. (21.03.2023) | Kranzer (2012) [39] | Cross-sectional (Outpatient) | Convenience sampling | Bacterial culture | Township population | Peri-urban areas, Cape Town | 1,011 | Adults | 64 | 47 |
| L.H.; A.J. (01.03.2023) | Cox (2010) [40] | Cross-sectional (Outpatient) | Convenience sampling | Bacterial culture | Township population | Khayelitsha, Cape Town | 1,630 | Adults | NR | 55–71* |
| L.H.; A.J. (28.02.2023) | Middelkoop (2010) [41] | Cross-sectional (Home) | Population-based sampling | Sputum smear microscopy Bacterial culture | Township population | Cape Town | 1,250 | Adults (≥ 15 years) | 48 | 25 |
| L.H.; A.J. (02.03.2023) | Van Rie (2018) [42] | Cross-sectional (Home) | Population-based sampling | Automated NAAT Self-report | Township population | Diepsloot, Johannesburg | 1,231 | Adults (≥ 15 years) | 54 | 8.4 (New diagnosis) |
| L.H.; A.J. (07.03.2023) | Yates (2018) [43] | Cross-sectional (Home) | Population-based sampling | Bacterial culture | Low SES† | 8 communities, Western Cape | 15,036 | Adults | NR | 9.9‡ |
| L.H.; A.J. (01.03.2023) | Govender (2010) [44] | Cross-sectional (Home) | Unclear | Self-report (Household survey) | Informal population | Low-cost housing communities, Cape Town | 370 | All ages | 50 | 3 |
| L.H.; A.J. (28.02.2023) | Cramm (2011) [45] | Cross-sectional (Home) | Population-based sampling | Self-report (Household survey) | Township population | Grahamstown, Eastern Cape | 977 | Adults | NR | NR |
| L.H.; A.J. (03.03.2023) | Booi (2022) [46] | Cross-sectional (Home) | Population-based sampling | Self-report (Household survey) | Township population | Mamelodi, Gauteng | 114,348 | NR | NR | NR |
| **b. LTBI PREVALENCE** | | | | | | | | | | |
| L.H.; A.J. (04.03.2023) | Ncayiyana. (2015) [47] | Cross-sectional (Home) | Population-based sampling | TST (≥5mm in PLWH, ≥10mm in others) | Township population | Diepsloot, Johannesburg | 446 | All ages | 60 | 18 |
| L.H.; A.J. (01.03.2023) | Wood (2010) [48] | Cross-sectional (Outpatient) | Population-based sampling | TST (≥10mm) | Township population | Cape Town | 1,061 | 5–17 (78); 18–40 years (22) | NR | 0 |
| L.H.; A.J. (02.03.2023) | Middelkoop (2014) [49] | Cross-sectional (Outpatient) | Population-based sampling | TST (≥10mm) | Township population | Cape Town | 1,100 | 5–22 years | 50 | 0 |
| L.H.; A.J. (04.03.2023) | Du Preez (2011) [50] | Cross-sectional (Home) | Convenience sampling | TST (≥10mm) | Township population | Uitsig/ Ravensmead, Cape Town | 196 | 3 months-15 years | 48 | 0 |
| L.H.; A.J. (02.03.2023) | Bunyasi (2019) [51] | Cross-sectional (Outpatient) | Convenience sampling | IGRA | Low SES§ | Cape Town | 5,929 | 12–19 years | NR | NR |
| **c. TB INCIDENCE** | | | | | | | | | | |
| L.H.; A.J. (03.03.2023) | Gupta (2012) [52] | Prospective cohort (Outpatient) | Convenience sampling | Various diagnostic tools (Incl. bacterial culture) | Township population | Gugulethu Township, Cape Town | 1,544 | Adults (≥ 16 years) | 70 | 100 |

(*Continued*)

**Table 1.** (Continued)

| Data extractors (Date) | Author (Year) | Study design (Study setting) | Sampling strategy | Diagnostic tool | Exposure definition | Geographic location | Sample size | Age group | Female (%) | PWH (%) |
|---|---|---|---|---|---|---|---|---|---|---|
| **a. TB PREVALENCE** | | | | | | | | | | |
| L.H.; A.J. (02.03.2023) | Martinez (2017) [53] | Prospective cohort (Outpatient) | Population-based sampling | Various diagnostic tools | Township population | Paarl, Cape Town | 915 | Children (Birth-5 years) | 49 | <1 |
| L.H.; A.J. (03.03.2023) | Wood (2010) [54] | Retrospective cohort (Outpatient) | Convenience sampling | Various diagnostic tools (Incl. bacterial culture and sputum smear microscopy) | Township population | Cape Town | 14,788 | All ages | NR | NR |
| L.H.; A.J. (09.08.2023) | Naidoo (2014) [55] | Prospective cohort (Outpatient) | Convenience sampling | Various diagnostic tools (Incl. bacterial culture) | Township population | Vulindlela, KwaZulu-Natal | 969 | Adults | 68 | 100 |
| L.H.; A.J. (07.03.2023) | Ilunga (2020) [56] | Prospective cohort (Home) | Population-based sampling | Self-report | Township population | Mamelodi, Gauteng | 184,351 | All ages | NR | NR |
| **d. LTBI INCIDENCE** | | | | | | | | | | |
| L.H.; A.J. (02.03.2023) | Middelkoop (2014) [49] | Retrospective cohort (Outpatient) | Population-based sampling | TST (≥10mm) | Township population | Cape Town | 67 | 5–22 years | 51 | 0 |
| L.H.; A.J. (02.03.2023) | Martinez (2017) [53] | Prospective cohort (Outpatient) | Population-based sampling | TST (≥10mm) | Township population | Paarl, Cape Town | 915 | Birth-5 years | 49 | <1 |

**Abbreviations:** TB = Tuberculosis; PLWH = People Living With HIV; NAAT = Nucleic Acid Amplification Test; NR = Not reported; LTBI = Latent Tuberculosis; TST = Tuberculin Skin Test; IGRA = Interferon-Gamma Release Assay.

*Among newly diagnosed TB cases and previously treated people with TB, respectively.

†Defined as individuals with a very low or low Household Wealth Index.

‡Among the overall population (individuals of all socio-economic statuses).

§Defined as individuals attending low-income state schools.

using self-report (n = 4) [43–45] tended to yield lower prevalence estimates than studies using bacterial culture (n = 8) [35–41, 43]. Furthermore, studies conducted in PLWH (n = 4) [35–38] tended to have higher active TB prevalence estimates than other studies. Lastly, four studies reported TB prevalence stratified by sex, ranging from 3.4% [42] to 34.8% [35] in women and from 6.1% [42] to 29.2% [35] in men.

**Incidence of active TB disease and LTBI.** Six studies reported active TB disease incidence estimates [52–56] (Table 4). These ranged from 0.7 [53] to 7.44 [52] people with TB per 100 person-years (n = 4 studies) [52–55]. The cumulative incidence ranged from 0.43% [56] to 31.4% [52] (n = 3 studies) [52, 54, 56]. One study reported sex-stratified TB incidence rates, finding a TB incidence of 2.4 people with TB per 100 person-years in women and 3.5 people with TB per 100 person-years in men [53]. Because of the small number of included studies, the use of different measures and units of incidence, and inconsistent age groupings, we could not assess the influence of age, HIV status, or diagnostic tool on TB incidence estimates. Of the two LTBI incidence studies, one was conducted in children and reported an incidence rate of 11.8 people with TB per 100 person-years overall, 9.4 people with TB per 100 person-years in girls, and 14.3 people with TB per 100 person-years in boys using TST [53]. The other study was conducted in children and adolescents and reported a cumulative incidence of 23.9% [49].

**Table 2. Risk-of-bias assessment of included studies.**

| Author (Year) | 1| Sample frame | 2| Sampling | 3| Sample size | 4| Study subjects and setting | 5| Data analysis | 6| Identification of the condition | 7| Measurement of the condition | 8| Statistical analysis | 9| Response rate | Overall appraisal |
|---|---|---|---|---|---|---|---|---|---|---|
| **a. TB PREVALENCE** | | | | | | | | | | |
| Cox (2010) [40] | No | No | Unclear | Yes | Yes | Yes | Yes | No | Yes | High risk of bias |
| Dawson (2010) [38] | No | No | Unclear | Yes | No | Yes | Yes | Yes | No | High risk of bias |
| Govender. (2010) [44] | Yes | Unclear | Unclear | Yes | Yes | No | Yes | No | Yes | High risk of bias |
| Middelkoop (2010) [41] | Yes | Yes | Unclear | Yes | Yes | Yes | Unclear | No | Yes | Low risk of bias |
| Cramm (2011) [45] | Yes | Yes | Unclear | No | Yes | No | Unclear | No | Yes | High risk of bias |
| Lawn (2011) [35] | No | No | Unclear | Yes | Yes | Yes | Yes | Yes | Yes | High risk of bias |
| Lawn (2011) [36] | No | No | Unclear | Yes | Yes | Yes | Yes | Yes | Yes | High risk of bias |
| Kranzer (2012) [39] | Yes | No | Unclear | Yes | No | Yes | Yes | Yes | No | High risk of bias |
| Lawn (2017) [37] | No | No | Unclear | Yes | Yes | Yes | Yes | Yes | Yes | High risk of bias |
| Van Rie (2018) [42] | Yes | Yes | Unclear | Yes | Yes | No | Yes | No | Yes | High risk of bias |
| Yates (2018) [43] | No | Yes | Unclear | Yes | No | Yes | Unclear | No | No | High risk of bias |
| Booi (2022) [46] | Yes | Yes | Yes | No | Yes | No | Unclear | No | Unclear | High risk of bias |
| **b. LTBI PREVALENCE** | | | | | | | | | | |
| Wood (2010) [48] | Yes | Yes | Unclear | No | NA | Yes | Unclear | Yes | NA | Low risk of bias |
| Du Preez (2011) [50] | No | No | Unclear | Yes | Unclear | Yes | Yes | No | Unclear | High risk of bias |
| Middelkoop (2014) [49] | Yes | Yes | Unclear | Yes | Unclear | Yes | Yes | Yes | Yes | Low risk of bias |
| Ncayiyana (2015) [47] | Yes | Yes | Unclear | Yes | Yes | Yes | Yes | Yes | Unclear | Low risk of bias |
| Bunyasi (2019) [51] | No | No | Yes | Yes | Unclear | Yes | Yes | Yes | Unclear | Exclude |
| **c. TB INCIDENCE** | | | | | | | | | | |
| Wood (2010) [54] | Yes | No | Unclear | Yes | Unclear | Yes | Unclear | Yes | Unclear | High risk of bias |
| Gupta (2012) [52] | No | No | Unclear | Yes | Unclear | Yes | No | Yes | Yes | High risk of bias |
| Naidoo (2014) [55] | No | No | Unclear | Yes | No | No | Yes | Yes | Yes | High risk of bias |
| Martinez (2017) [53] | Yes | Yes | Unclear | Yes | Unclear | Yes | Yes | Yes | Yes | Low risk of bias |
| Ilunga (2020) [56] | Yes | Yes | Yes | Yes | Yes | No | Unclear | No | Unclear | High risk of bias |
| **d. LTBI PREVALENCE** | | | | | | | | | | |
| Middelkoop (2014) [49] | Yes | Yes | Unclear | Yes | Unclear | Yes | Yes | Yes | Yes | Low risk of bias |

*(Continued)*

**Table 2.** (Continued)

| Author (Year) | 1\| Sample frame | 2\| Sampling | 3\| Sample size | 4\| Study subjects and setting | 5\| Data analysis | 6\| Identification of the condition | 7\| Measurement of the condition | 8\| Statistical analysis | 9\| Response rate | Overall appraisal |
|---|---|---|---|---|---|---|---|---|---|---|
| *Martinez (2017) [53]* | Yes | Yes | Unclear | Yes | Unclear | Yes | Yes | Yes | Yes | **Low risk of bias** |

**Abbreviations**: TB = Tuberculosis; LTBI = Latent Tuberculosis; NA = Not applicable.

*Because of the importance of sample frame (item 1), sampling approach (item 2) and diagnostic methods (item 6) for determining prevalence and incidence in a target population, studies receiving a 'No" response to any of these items were labelled as 'high risk-of-bias'.

## Pooled results

**Primary analysis.** A meta-analysis of eight eligible studies [35–41, 43] yielded a pooled active TB prevalence of 15.4% (95% CI 7.7–25.3) among PLWH, 2.7% (95% CI 0.1–8.5) among those living without HIV, and 7.9% (95% CI 0.4–23) for studies including both PLWH and those living without HIV (*S1 Fig*). The $I^2$ was >95% in all three groups. The small number of eligible studies prevented us from making conclusions about publication bias for any of the subgroups, although the Egger test failed to reject the null hypothesis of symmetry (*S2–S4 Figs, S5 Table*). Meta-analysis of four eligible studies [47–50] yielded a pooled LTBI prevalence of 43.4% (95% CI 39.5–47.3) in people living without HIV, with an $I^2$ of 71.6% (*S5 Fig*). LTBI prevalence was 33% (95% CI 22.6–44.4) in the one study conducted among PLWH [47] and 55% (95% CI 53.3–55.9) in the one study including both PLWH and those living without HIV [51]. The Egger test for asymmetry was not significant (*S5 Table*). Due to the small number of eligible studies and their heterogeneity, we could not draw any conclusions about publication bias from the funnel plot (*S6 Fig*). Additionally, because of the insufficient number of qualifying studies reporting summary estimates using consistent incidence units (<2 within each meta-analysis group), we did not conduct meta-analyses for active TB disease incidence or for LTBI incidence.

**Sensitivity and subgroup analyses.** After removing two studies [39, 41] that were major contributors to overall heterogeneity because of differences in study design and participant characteristics, meta-analysis of the remaining four studies yielded a pooled active TB prevalence of 22.7% (95% CI 15.8–30.4) among PLWH. In the group of studies including both people living with and without HIV, we excluded one study [40] that was a statistically significant outlier (Cook's distance = 1.1). The remaining three studies had a pooled active TB prevalence of 2.9% (95% CI 1.5–4.8; *Fig 2*). The $I^2$ remained high at > 90%. Excluding one study [47], a statistically significant outlier (Cook's distance = 0.47), from the studies including people living without HIV resulted in a pooled LTBI prevalence of 44.8% (95% CI 42.5–47) in the remaining three studies, with the $I^2$ reduced to 14.6% (*Fig 3*). *S6 Table* compares pooled prevalence estimates obtained before and after excluding outliers.

We were unable to conduct formal subgroup analyses by potential drivers of heterogeneity such as diagnostic tool, age, sex, ethnicity, or study setting used because these variables were only reported in <10 studies or reported inconsistently (e.g., differing age categories) [25, 26]. We were also unable to conduct sensitivity analyses based on the risk-of-bias assessment results because only one study reporting TB prevalence and only three studies reporting LTBI prevalence were considered to have a low risk of bias.

**Prevalence ratios for active TB disease and LTBI relative to the general population.** People living without HIV in underserved populations had an almost 4-fold greater risk of active TB disease (PR = 3.8) than the overall South African population. Populations including

**Table 3. Prevalence of active TB disease and LTBI in included studies.**

| a. TB PREVALENCE | | | | | | |
|---|---|---|---|---|---|---|
| Author (Year) | HIV status | Sample size | TB cases (% Prevalence) | Diagnostic tool | Data collection unit | Included in meta-analysis? (Reason for exclusion) |
| *Lawn (2011) [35]* | Positive | 468 | 81 (17.3) | Bacterial culture | Individual | Yes (Not applicable) |
| *Lawn (2017) [36]* | Positive | 427 | 139 (32.6) | NAAT; Bacterial culture | Individual | Yes (Not applicable) |
| *Lawn (2011) [37]* | Positive | 542 | 94 (17.3) | Bacterial culture | Individual | Yes (Not applicable) |
| *Dawson (2010) [38]* | Positive | 235 | 58 (24.7) | Bacterial culture | Individual | Yes (Not applicable) |
| *Kranzer (2012) [39]* | Mixed† | 1 011 | 56 (5.5) | Bacterial culture | Individual | Yes (Not applicable) |
| | Positive | 520 | 30 (5.8) | | | |
| | Negative | 491 | 26 (5.3) | | | |
| *Cox (2010) [40]* | Mixed† | 1 575 | 535 (34) | Bacterial culture | Individual | Yes (Not applicable) |
| | Positive | NR | 300 | | | |
| | Negative | NR | 176 | | | |
| | NR | NR | 59 | | | |
| *Middelkoop (2010) [41]* | Mixed† | 1 250 | 20 (1.6) | Sputum smear-microscopy; Bacterial culture | Individual | Yes (Not applicable) |
| | Positive | 306 | 11 (3.6) | | | |
| | Negative | 901 | 9 (1) | | | |
| | NR | 43 | NR | | | |
| *Van Rie (2018) [42]* | Mixed† | 1 231 | NAAT*: 5 (0.4) Self-report: 57 (4.6) | NAAT: Self-report | Individual | No (Use of self-report) |
| *Yates (2018) [43]* | Mixed† | 15 036 | 371 (2.5) | Bacterial culture | Individual | Yes (Not applicable) |
| *Govender (2010) [44]* | Mixed† | 370 | 14 (3.8) | Self-report | Household | No (Use of self-report) |
| *Cramm (2011) [45]* | NR | 977 | 316 (32.5) | Self-report | Household | No (Use of self-report) |
| *Booi (2022) [46]* | NR | 114 348 | 1 742 (1.5) | Self-report | Individual | No (Use of self-report) |
| b. LTBI PREVALENCE | | | | | | |
| *Ncayiyana (2015) [47]* | Mixed† | 446 | 153 (34.3) | TST | Individual | Yes (Not applicable) |
| | Positive | 70 | 23 (32.9) | | | |
| | Negative | 317 | 115 (36.3) | | | |
| | NR | 59 | 15 (25.4) | | | |
| *Wood (2010) [48]* | Negative | 1 061 | 477 (45) | TST | Individual | Yes (Not applicable) |
| *Middelkoop (2014) [49]* | Negative | 1 100 | 480 (43.6) | TST | Individual | Yes (Not applicable) |
| *Du Preez (2011) [50]* | Negative | 196 | 97 (49.5) | TST | Individual | Yes (Not applicable) |
| *Bunyasi (2019) [51]* | NR | 5 929 | 3 236 (54.6) | IGRA | Individual | Yes (Not applicable) |

**Abbreviations**: TB = Tuberculosis; NAAT = Nucleic Acid Amplification Assay; NR = Not reported; LTBI = Latent Tuberculosis; TST = Tuberculin Skin Test; IGRA = Interferon-Gamma Release Assay.

*NAAT only performed in individuals with presumptive TB.

†Including both people living with HIV and people living without HIV (the overall study cohort) when study outcomes were not broken down by HIV status.

‡Sex-stratified outcome estimates are not provided due to the limited number of studies reporting Tb/LTBI prevalence separately for each sex.

Five studies reported on LTBI prevalence [47–51], ranging from 25.4% [47] to 54.6% [51]. Four studies sampled pediatric and adolescent populations [48–51], with LTBI prevalence ranging from 43.6% [49] to 54.6 [51]. The three studies among populations living without HIV [48–50] showed comparable LTBI prevalence estimates as the studies conducted among PLWH. One study reported LTBI prevalence by sex, finding a LTBI prevalence of 32.3% in women and 37.1% in men [47].

both PLWH and people living without HIV also had an almost 4-fold higher risk (PR = 3.9) of active TB disease. PLWH were at a 31-fold increased risk (PR = 30.7). For LTBI, we predicted people living without HIV to be at a 3.3-fold increased risk compared to the overall population. Based on a single study estimate, we estimated a PR of 4 in a population with both PLWH and

**Table 4. Incidence of active TB disease and LTBI in included studies.**

| a. TB INCIDENCE | | | | | | |
|---|---|---|---|---|---|---|
| Author (Year) | HIV status | Sample size | TB cases (% Incidence) | TB incidence rate [95%CI] | Diagnostic tool | Data collection unit |
| *Gupta (2012)* [52] | Positive | 1 544 | Total: 484 (31.4) Culture-confirmed: 356 (23.1) | 7.44 [6.8–8.13] people with TB per 100 PY Culture-confirmed: 23 057 [21 762–24 288] people with TB per 100 000 | Various | Individual |
| *Martinez (2017)* [53] | Negative* | 915 | Total: 81 (8.9) Microbiologically confirmed: 18 (2) | Total: 2.9 [2.4–3.7] per 100 PY Microbiologically confirmed: 0.7 [0.4–1.0] per 100 PY | Various | Individual |
| Wood (2010) [54] | NR | Total: 14 788 Adults: 12 097 Children (5–15): 1 640 Children (<5): 1 051 | Total: 1 289 (8.7) Adults: 670 (5.5) Children (5–15): 45 (2.7) Children (<5): 86 (8.2) | Total population: 1909 [1799–2018] per 100 000 PY Adults: (culture-confirmed): 1347 [1 108–1 437] per 100 000 PY Adults (culture-confirmed): 5 539 people with TB per 100 000 Children (5–15): 546 [346–546] per 100 000 PY Children (<5): 1522 [1419–1533] per 100 000 PY | Various | Individual |
| *Naidoo (2014)* [55] | positive | 969 | 54 (5.6) | 4.5 [3.3–5.8] cases per 100 PY | Various | Individual |
| *Ilunga (2010)* [56] | NR | 184 351 | 788 (0.4) | 427 people with TB per 100 000 | Self-report | Individual |
| b. LTBI INCIDENCE | | | | | | |
| *Middelkoop (2014)* [49] | Negative | 67 | 16 (23.9) | 23 881 people with TB per 100 000 | TST | Individual |
| *Martinez (2017)* [53] | Negative* | 915 | 147 (16) | 11.8 [10–13.8] per 100 PY | TST | Individual |

**Abbreviations**: TB = Tuberculosis; PY = Person-years; NR = Not reported; LTBI = Latent Tuberculosis; TST = Tuberculin Skin Test.

*The 2 (<1) children living with HIV were not separately analysed and included in the cohort of children without HIV; all TB incidents occurred in children living without HIV.

†Sex-stratified outcome estimates are not provided due to the limited number of studies reporting TB/LTBI prevalence separately for each sex.

those living without HIV [51]. The prevalence estimate for LTBI in the one study of PLWH [47] was almost the same as the national estimate, yielding a PR of 1.

## Discussion

In this updated systematic review of the literature published between 2010–2023, we found that underserved populations living without HIV had an almost 4-fold increased risk of active TB disease and a 3.3-fold increased risk of LTBI compared to the general population in South Africa. The risk of active TB was even greater in underserved PLWH. Our findings are, however, limited by substantial heterogeneity among a small number of studies distributed over a long period of time during which TB and HIV policies and practices have been rapidly changing. In addition, the generally low quality of study reporting further limits our confidence in these results.

Our findings are in line with previous research conducted among underserved populations worldwide. A prior systematic review conducted in South Africa and including studies published between 2000 and 2011 reported a 5.8-fold increased risk of active TB disease in informal settlements [14]. In contrast, a modelling study on population-level risk factors for TB in South Africa in 2010 failed to find a significantly higher active TB disease risk among those living in informal settlements [57]. However, because TB prevalence was derived from laboratory reports of detected TB incidents as a proxy for prevalence and not from a population-based sample, the true TB disease burden in informal settlements was likely substantially

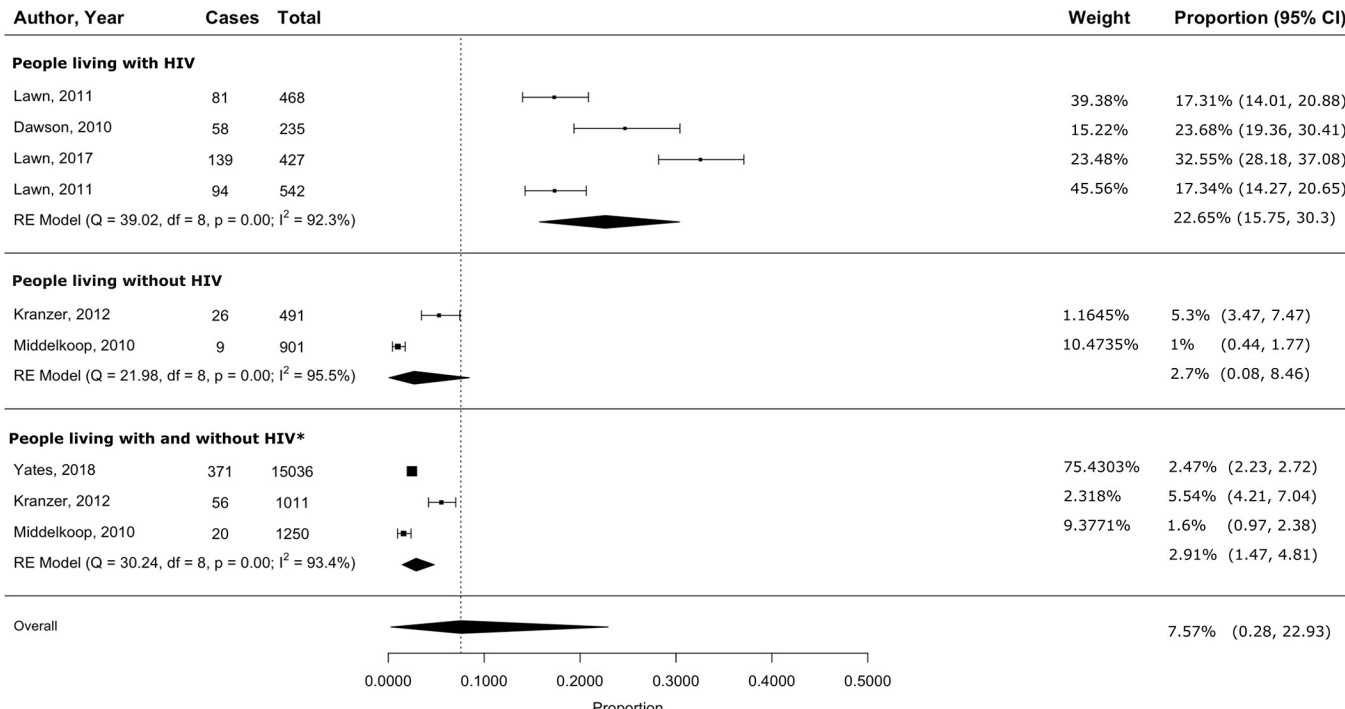

**Fig 2. Pooled active TB disease prevalence among underserved populations in South Africa, stratified by HIV status. Abbreviations**: HIV = Human Immunodeficiency Virus. *The 'People living with and without HIV' group includes studies for which outcomes were not stratified by HIV status.

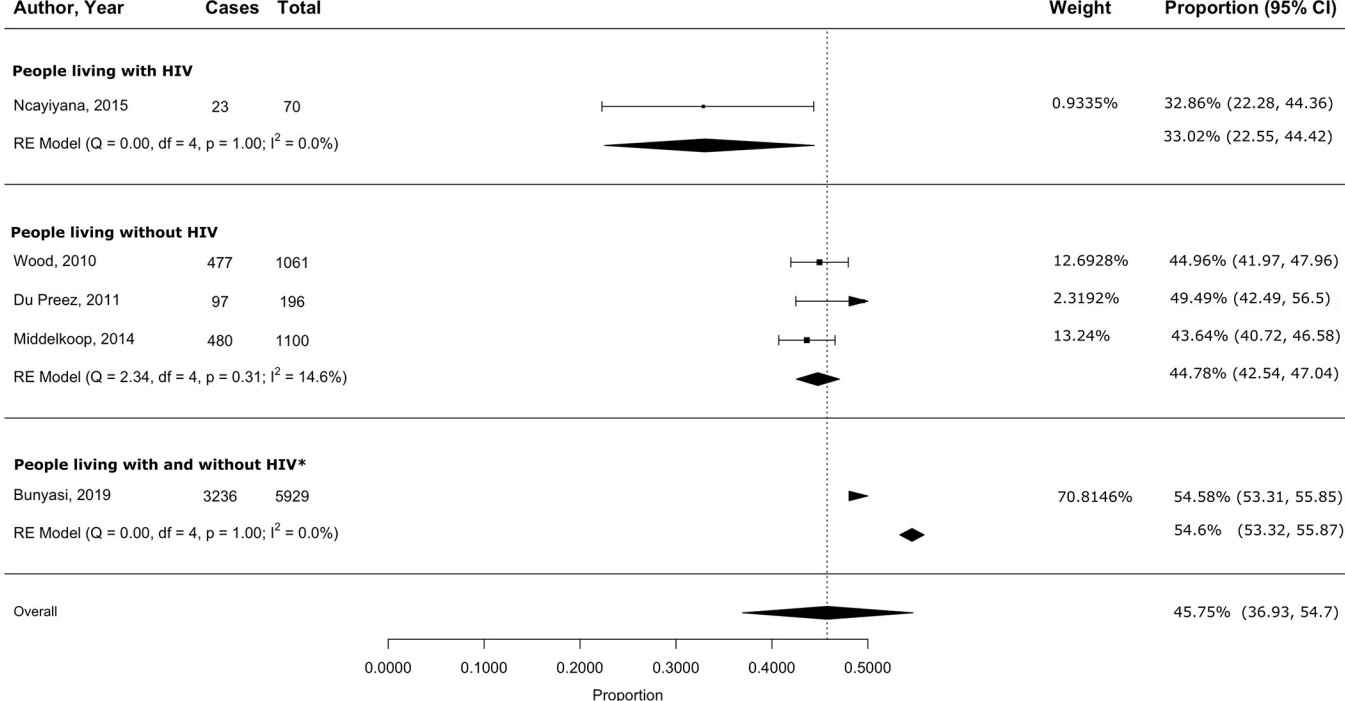

**Fig 3. Pooled LTBI prevalence among underserved populations in South Africa, stratified by HIV status. Abbreviations**: HIV = Human Immunodeficiency Virus. *The 'People living with and without HIV' group includes studies for which outcomes were not stratified by HIV status.

underestimated. An international review of the TB risk in slum households, an alternative term for those living in informal settlements, also found that the incidence of smear-positive TB in slums was 2.96 times higher than the national TB incidence [58]. Likewise, a population-based, cross-sectional study conducted in slum settings in Uganda in 2019 found that the TB prevalence was four times the national estimate [59]. Finally, a prospective implementation study of active TB case finding in Nigerian slums in 2012 yielded a TB prevalence twice that of the national average, of which 22.6% were living with HIV [60].

There were several limitations to the studies included in this review. Firstly, most studies were not designed primarily to measure incidence and prevalence, but to address other research questions. As a result, they were limited by small sample sizes and non-population-based sampling, leading to uncertain and biased estimates of the true TB and LTBI prevalence among underserved populations in South Africa. Conducting population-based studies in resource-limited settings is particularly challenging given constraints in diagnostic availability and associated costs [61, 62]. Consequently, we identified only one low risk-of-bias study among the studies reporting TB prevalence that employed population-based sampling [41]. A recent comprehensive overview of systematic reviews on TB prevalence and incidence in underserved populations worldwide also underscored deficiencies in the quality of included studies, as assessed by the same risk-of-bias assessment tool employed in our review [12]. Secondly, our broad definition of underserved populations, the long study inclusion period, demographic differences, and diverse diagnostic tools across studies may have introduced heterogeneity, as reflected in the high $I^2$ values in the pooled prevalence estimates. Unfortunately, the limited number of studies prevented more detailed subgroup analyses to explore heterogeneity. Subgroup analyses based on sex and age would have been particularly relevant due to significant variations in LTBI prevalence across different age groups and between men and women [31]. Two out of four studies that reported sex-stratified TB prevalence estimates found slightly higher numbers in women than in men. This contrasts with nationwide estimates, which show higher TB prevalence and mortality in men than in women [63]. All other studies that provided sex-specific outcomes reported lower prevalence and incidence estimates of active TB disease and LTBI in women compared to men. Thirdly, assessing publication bias was challenging due to the limited power of funnel plots and the Egger test with small study numbers [26]. Furthermore, the high concentration of study samples from around Cape Town limits the generalisability of our findings, emphasising the need for additional research from other cities and provinces, including rural areas to provide a more representative view of TB epidemiology in South Africa. Further research is needed not only in different geographical areas but also among other at-risk populations, including but not limited to rural poor communities, people who are incarcerated, and refugee and migrant populations [14]. Also, few studies investigated other risk factors like diabetes mellitus, tobacco smoking, alcohol use, or malnutrition in relation to overall TB risk. Given their high prevalence in socioeconomically disadvantaged populations and potential impacts on pathogenesis and treatment, quantifying their prevalence in individuals with TB would be beneficial. Lastly, data on the impacts of the COVID-19 pandemic on the TB burden in underserved populations is scarce [64]. These populations face increased vulnerability to the pandemic's consequences due to a loss of household income without a financial safety net and poor access to social assistance programmes, testing, and healthcare services [65].

Despite the rigorous methodology employed in this review, there are several weaknesses. First, the risk for LTBI among PLWH appears comparable to that of the general population in the primary analysis, as indicated by a single study estimate. However, this finding should be interpreted with caution, as the prevalence of LTBI remains higher among PLWH compared to those living without HIV. This is despite the implementation of TB preventive therapy

among household contacts of infectious pulmonary TB and TST-confirmed LTBI cases, for which uptake remains poor [66, 67]. The credibility of this finding is further diminished by the low sensitivity of both the TST and IGRA in detecting LTBI among PLWH [66]. Second, we used the most up-to-date 2018 national TB prevalence estimate as the denominator for active TB prevalence ratios [30]. However, because many studies contributing to the numerator were published around 2010 when TB prevalence was higher, the 2018 national estimate likely underestimated the national TB prevalence in earlier years, potentially inflating our estimates of the increased TB risk in underserved populations. Further, the use of modelled national LTBI estimates from 2014 as the denominator for LTBI prevalence ratios introduces similar limitations [31]. Two of the three studies contributing to the numerator including people without HIV—the only subgroup with more than one study–were published in 2010 and 2011, during a period when TB prevalence was higher than in 2014. This may have resulted in an overestimation of the LTBI risk in underserved people without HIV. Finally, the risk-of-bias assessment tool we used was designed for prevalence studies and does not consider the importance of length of follow-up to obtain valid cumulative incidence estimates [23].

Our study also has several strengths. First, this review employed a rigorous and systematic search of both South African and international databases. This is reflected in the substantial number of studies incorporated into our review, the majority of which were not captured in the previously conducted scoping review on TB prevalence and incidence in informal settlements in South Africa [14]. This suggests a significant expansion in the pool of available studies since then, and that our search strategy successfully captured a broad spectrum of studies. Moreover, to enhance precision in our pooled prevalence estimates, our meta-analysis only incorporated prevalence estimates obtained using WHO-approved diagnostic tools. Other review strengths are the use of a risk-of-bias assessment tool specifically tailored for prevalence studies, and the thorough review process, with two independent reviewers conducting all stages of study screening, data extraction and risk-of-bias assessment.

The results from our analysis have contributed to updated TB prevalence estimates for people living in informal settlements in South Africa's National TB Strategic Plan 2023–2028 [5]. Their population size has been steadily growing over the past 20 years in South Africa and many other countries worldwide, and is projected to further increase in the coming decades [68, 69]. Consequently, informal settlements are expected to remain a significant contributor to both the global TB prevalence and incidence. To effectively curb these figures, it is imperative to implement a diverse array of testing modalities that are acceptable, feasible, cost-effective, and can be seamlessly integrated into targeted case-finding initiatives that focus on 'hard-to-reach' populations [70–73]. For example, a community-based active case-finding initiative using rapid point-of-care (POC) GeneXpert testing in mobile clinics within peri-urban informal settlements of Cape Town has shown promise, by decreasing time to treatment initiation and increasing the percentage of people receiving TB treatment compared to facility-based testing using sputum smear microscopy [74]. Strategies like these may be especially beneficial in informal settings where laboratory facilities and qualified medical personnel are scarce, by helping reach individuals who do not present to healthcare facilities [75, 76]. The National TB Strategic Plan 2023–2028 identifies informal settlements as a priority population for novel POC diagnostic tools, such as digital chest x-ray, tongue swabs, and self-screening [5]. Additionally, it notes that treatment modalities like community-based care and adherence support should be considered, bolstered by laboratory data and geo-spatial mapping [5]. In addition, the expansion of existing surveillance systems plays a pivotal role in evaluating the effectiveness of targeted TB interventions. In the long-term, changes to urban infrastructure and integration of healthcare services are needed to sustainably decrease TB prevalence and promote health equity. Importantly, this cannot be achieved without addressing the enduring economic

and social repercussions of apartheid, which continue to perpetuate racial inequities in housing and other social determinants of health in South Africa [77].

## Conclusions

This review adds to a pool of evidence that emphasises the key contribution of underserved populations to the South African TB epidemic. Thus, this review can serve as an important resource for national stakeholders and TB programs in assessing the relative contributions of various populations to the TB epidemic in South Africa and other TB-endemic settings.

## Supporting information

**S1 Checklist. PRISMA 2020 checklist [1].**
(DOCX)

**S1 Table. Search term.**
(DOCX)

**S2 Table. Data extraction form.**
(DOCX)

**S3 Table. Risk-of-bias assessment form [2].**
(DOCX)

**S4 Table.**
(DOCX)

**S5 Table. Egger tests: P-values.**
(DOCX)

**S6 Table. Pooled prevalence before and after sensitivity analysis.**
(DOCX)

**S1 Data. Sensitivity analysis.**
(DOCX)

**S2 Data. Prevalence ratios.**
(DOCX)

**S3 Data. Screened studies, broken down into excluded and included studies.**
(XLSX)

**S1 Fig. Pooled active TB disease prevalence among underserved populations in South Africa, stratified by HIV status. Abbreviations:** HIV = Human Immunodeficiency Virus. *The 'People living with and without HIV' group includes studies for which outcomes were not stratified by HIV status.
(DOCX)

**S2 Fig. Funnel plot: Pooled active TB disease prevalence among underserved populations in South Africa ('People living with HIV' subgroup). Abbreviations:** TB = Tuberculosis; HIV = Human Immunodeficiency Virus.
(DOCX)

**S3 Fig. Funnel plot: Pooled active TB disease prevalence among underserved populations in South Africa ('People living without HIV' subgroup). Abbreviations:** TB = Tuberculosis;

HIV = Human Immunodeficiency Virus.
(DOCX)

**S4 Fig. Funnel plot: Pooled active TB disease prevalence among underserved populations in South Africa ('People living with and without HIV' subgroup). Abbreviations:** TB = Tuberculosis; HIV = Human Immunodeficiency Virus.
(DOCX)

**S5 Fig. Pooled LTBI prevalence among underserved populations in South Africa, stratified by HIV status. Abbreviations:** HIV = Human Immunodeficiency Virus. *The 'People living with and without HIV' group includes studies for which outcomes were not stratified by HIV status.
(DOCX)

**S6 Fig. Funnel plot: Pooled LTBI prevalence among underserved populations in South Africa ('People living without HIV' subgroup). Abbreviations:** LTBI = Latent Tuberculosis; HIV = Human Immunodeficiency Virus.
(DOCX)

## Author Contributions

**Conceptualization:** Lydia M. L. Holtgrewe, Ann Johnson, Kate Nyhan, J. Lucian Davis, Salome Charalambous.

**Data curation:** Lydia M. L. Holtgrewe, Ann Johnson.

**Formal analysis:** Lydia M. L. Holtgrewe.

**Investigation:** Lydia M. L. Holtgrewe.

**Methodology:** Lydia M. L. Holtgrewe, Ann Johnson, Kate Nyhan, Jody Boffa, Sheela V. Shenoi, Aaron S. Karat, J. Lucian Davis.

**Project administration:** Lydia M. L. Holtgrewe, J. Lucian Davis, Salome Charalambous.

**Resources:** Lydia M. L. Holtgrewe, J. Lucian Davis.

**Software:** Lydia M. L. Holtgrewe, Kate Nyhan.

**Supervision:** J. Lucian Davis, Salome Charalambous.

**Validation:** J. Lucian Davis, Salome Charalambous.

**Visualization:** Lydia M. L. Holtgrewe.

**Writing – original draft:** Lydia M. L. Holtgrewe, J. Lucian Davis.

**Writing – review & editing:** Lydia M. L. Holtgrewe, Ann Johnson, Kate Nyhan, Jody Boffa, Sheela V. Shenoi, Aaron S. Karat, J. Lucian Davis, Salome Charalambous.

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
