## [Decision Letter · Decision Letter 0]

3 May 2024

PGPH-D-24-00621

Burden of tuberculosis in underserved populations in South Africa: A systematic review and meta-analysis

Dear Dr. Holtgrewe,

Thank you for submitting your manuscript to PLOS Global Public Health. After careful consideration, we feel that it has merit but does not fully meet PLOS Global Public Health’s publication criteria as it currently stands. Therefore, we invite you to submit a revised version of the manuscript that addresses the points raised during the review process.

While we appreciate the merit of the manuscript submitted, we would please request that the authors specfically address the following comments from the reviewers:

Based on the various criteria presented in tables S4, clarify how a study was considered low vs high quality and included or excluded from the review as per table S5.  Consider moving this information to the main manuscript as suggested by the reviewer.Define what were considered "ineligible measures of disease burden"

Further, with regards to WHO endorsed diagnostic tools (line 128-129): no mention is made of the LF-LAM assay.  Could the authors please clarify why this WHO endorsed diagnostic tool, and also recommended by South Africa NTP since 2021, was not included ? Particularly in the view that 6 studies included in this review focused on TB in PLWH only.

We look forward to receiving your revised manuscript.

Kind regards,

Marguerite Massinga Loembe, PhD

Academic Editor

Journal Requirements:

Additional Editor Comments (if provided):

Reviewers' comments:

Reviewer's Responses to Questions

**Comments to the Author**

1. Does this manuscript meet PLOS Global Public Health’s publication criteria? Is the manuscript technically sound, and do the data support the conclusions? The manuscript must describe methodologically and ethically rigorous research with conclusions that are appropriately drawn based on the data presented.

Reviewer #1: Yes

Reviewer #2: Yes

2. Has the statistical analysis been performed appropriately and rigorously?

Reviewer #1: Yes

Reviewer #2: Yes

3. Have the authors made all data underlying the findings in their manuscript fully available (please refer to the Data Availability Statement at the start of the manuscript PDF file)?

Reviewer #1: Yes

Reviewer #2: Yes

4. Is the manuscript presented in an intelligible fashion and written in standard English?

Reviewer #1: Yes

Reviewer #2: Yes

5. Review Comments to the Author

Reviewer #1: Thank you for the opportunity to review this interesting study. This is an important analysis that attempts to estimate the burden of tuberculosis (TB) among underserved populations in South Africa–a high TB burden country. In the systematic review, the authors identified 22 studies for inclusion and found that TB burden was substantially greater among underserved populations compared to the overall population. I have the following comments to improve the clarity of the study:

Introduction:

Lines 51–56: The authors mention the WHO End TB Targets. For readers who are unfamiliar, the authors should briefly define the targets.

Line 52: Consider also citing the latest Global Burden of Disease report illustrating that the End TB targets were not achieved at the global level.

https://pubmed.ncbi.nlm.nih.gov/38518787/

Please use PLWH for people living with HIV rather than PWH.

Methods:

Lines 86 – 88: Why wasn’t PubMed also queried? This is important because prior work has shown that a combination of Embase and PubMed yields high coverage.

https://pubmed.ncbi.nlm.nih.gov/29208034/

Study selection: Please include additional information for inclusion criteria such as types of study designs were included, sampling methods, populations, outcome measurement, etc. This is important because I’m very unclear on the exact inclusion criteria for the systematic review.

Quality assessment: Please include 2-3 sentences on which metrics were used to evaluate study quality and why. Please also reference Table S4 here to point to additional details for readers. In addition, what is considered to be high-quality vs low-quality?

Sensitivity analyses: Was it possible to conduct sub-group analyses by sex? It may be interesting to see if the male to female ratio among underserved population differs compared to other estimates.

Other effect estimates: Have the authors considered using estimates from the Global Burden of Disease study as the comparator for the overall national-level TB prevalence in South Africa?

http://ihmeuw.org/6e62

Results:

Lines 169: What are ineligible measures of disease burden?

Table 1: The authors mention that many studies are not population-based but recruited. This is an important point. Can the authors include an additional column in Table 1 for recruitment strategy/sampling?

Discussion:

Lines 367–376: Can the authors comment on previous interventions that have been shown to improve case finding and TB outcomes among underserved populations. Since the goal of the study was to help inform the national TB program in South Africa, this provide valuable insights.

Can the authors briefly comment how the COVID-19 pandemic may have impact TB burden among underserved populations in South Africa?

Reviewer #2: Summary: A systematic review and meta-analysis of studies undertaken to assess burden of TB ( active TB prevalence and incidence and LTBI prevalence and incidence ) in underserved populations in South Africa . The review found a high burden of TB in this population compared with the South African general population (four and 31 fold higher risk of active TB in those without HIV and those with HIV respectively).

Strengths: The review is of high quality and conforms with accepted norms for the conduct of a systematic review and meta-analysis including in methodological approach, literature search strategy, assessment of bias, measures/ascertainment of outcomes etc.

Weakness:

1. A description of the burden of TB in South Africa, including incidence and prevalence is not provided to allow readers of this paper to have a better understanding of the TB situation in this country. The introduction only includes a statement about TB treatment coverage in South Africa (line 52-55).

2. While the review was focused on underserved” populations including people living in informal settlements, townships and impoverished communities only the population of people living in informal settlements is clearly defined (Lines 101-103). It is not clear if living in informal settlement is the same as living in a township. The measure of impoverishment in South Africa is also not defined.

3. It is noted that included studies were mostly carried out among township populations ( 9 of 12 studies for prevalence of active TB, 4 out of 5 studies for LTBI prevalence, 5 of 5 studies for TB incidence and 2 of 2 studies for LTBI incidence) and mostly in Cape Town. While the authors acknowledge this limitation, there is no apparent push or argument to conduct research studies similar to those included in this systematic review in other regions of the country and also in other populations considered at risk for TB ( for example rural poor populations, people deprived of liberty and others). It is therefore uncertain how the results of this study will inform the development of South Africa's National TB Strategy.

4. The “global value” of the results obtained by this systematic review and meta-analysis appears limited – what this systematic review and meta-analysis reveals has been known for a long time.

6. PLOS authors have the option to publish the peer review history of their article (what does this mean?). If published, this will include your full peer review and any attached files.

**Do you want your identity to be public for this peer review?** For information about this choice, including consent withdrawal, please see our Privacy Policy.

Reviewer #1: No

Reviewer #2: **Yes: **Jeremiah Chakaya

---

## [Decision Letter · Decision Letter 1]

29 Aug 2024

PGPH-D-24-00621R1

Burden of tuberculosis in underserved populations in South Africa: A systematic review and meta-analysis

Dear Dr. Holtgrewe,

Thank you for submitting your revised manuscript to PLOS Global Public Health and for comprehensively responding to the reviewers' comments. After careful consideration, two (2) minor comments would need to be addressed. Therefore, we invite you to submit a revised version of the manuscript that addresses these pending points raised during the second review process.

We look forward to receiving your revised manuscript.

Kind regards,

Marguerite Massinga Loembe, PhD

Academic Editor

Journal Requirements:

Additional Editor Comments (if provided):

Line 424 of the revised manuscript with track changes:

Authors cite the article by Kubjane, M., Cornell, M., Osman, M., Boulle, A., & Johnson, L. F. (reference 63).  This paper indicated that "the M:F ratios for tuberculosis incidence and mortality rates persisted above 1.0, and the figures reached 1.70 and 1.65, respectively, by the end of 2019" and further that "the 2019 estimated tuberculosis prevalence in males was 1.06% (95% CI 1.0–1.12%) and 0.58% (95% CI 0.56–0.62%) in females".

These observations appear to be contradicting the authors' statement on line 424 that "..nationwide estimates, ... show higher TB prevalence and mortality in women than in men".  Could this please be cross checked for consistency ?

Reviewers' comments:

Reviewer's Responses to Questions

**Comments to the Author**

1. If the authors have adequately addressed your comments raised in a previous round of review and you feel that this manuscript is now acceptable for publication, you may indicate that here to bypass the “Comments to the Author” section, enter your conflict of interest statement in the “Confidential to Editor” section, and submit your "Accept" recommendation.

Reviewer #1: All comments have been addressed

Reviewer #2: All comments have been addressed

2. Does this manuscript meet PLOS Global Public Health’s publication criteria? Is the manuscript technically sound, and do the data support the conclusions? The manuscript must describe methodologically and ethically rigorous research with conclusions that are appropriately drawn based on the data presented.

Reviewer #1: Yes

Reviewer #2: Yes

3. Has the statistical analysis been performed appropriately and rigorously?

Reviewer #1: Yes

Reviewer #2: Yes

4. Have the authors made all data underlying the findings in their manuscript fully available (please refer to the Data Availability Statement at the start of the manuscript PDF file)?

Reviewer #1: Yes

Reviewer #2: Yes

5. Is the manuscript presented in an intelligible fashion and written in standard English?

Reviewer #1: Yes

Reviewer #2: Yes

6. Review Comments to the Author

Reviewer #1: I thank the authors for their diligent edits to the manuscript. The authors have addressed all my comments. The paper provides a valuable contribution to understanding the burden of TB in underserved populations in a high TB burden country.

Reviewer #2: The revised manuscript has provided satisfactory responses to the comments made in the previous review. I have only a very minor comment. On page 7 line 54 it is stated that TB treatment coverage (TC) increased from 57% to 77% in 2022. Could you please provide the year when the the TB treatment coverage was 57% - is the baseline year 2015?

7. PLOS authors have the option to publish the peer review history of their article (what does this mean?). If published, this will include your full peer review and any attached files.

**Do you want your identity to be public for this peer review?** For information about this choice, including consent withdrawal, please see our Privacy Policy.

Reviewer #1: No

Reviewer #2: No

---

## [Editor Report · Decision Letter 2]

3 Sep 2024

Burden of tuberculosis in underserved populations in South Africa: A systematic review and meta-analysis

PGPH-D-24-00621R2

Dear Miss Holtgrewe,

We are pleased to inform you that your manuscript 'Burden of tuberculosis in underserved populations in South Africa: A systematic review and meta-analysis' has been provisionally accepted for publication in PLOS Global Public Health.

Best regards,

Marguerite Massinga Loembe, PhD

Academic Editor